# Interleukin-21 in Viral Infections

**DOI:** 10.3390/ijms22179521

**Published:** 2021-09-01

**Authors:** Hironobu Asao

**Affiliations:** Department of Immunology, Faculty of Medicine, Yamagata University, 2-2-2 Iida-nishi, Yamagata City 990-9585, Japan; asao-h@med.id.yamagata-u.ac.jp

**Keywords:** interleukin (IL)-21, CD4^+^ T cell, follicular helper T (Tfh) cell, B cell, CD8^+^ T cell, T cell exhaustion

## Abstract

Interleukin (IL)-21 is a cytokine that affects the differentiation and function of lymphoid and myeloid cells and regulates both innate and adaptive immune responses. In addition to regulating the immune response to tumor and viral infections, IL-21 also has a profound effect on the development of autoimmune and inflammatory diseases. IL-21 is produced mainly from CD4^+^ T cells—in particular, follicular helper T (Tfh) cells—which have a great influence on the regulation of antibody production. It is also an important cytokine for the activation of CD8^+^ T cells, and its role in recovering the function of CD8^+^ T cells exhausted by chronic microbial infections and cancer has been clarified. Thus, IL-21 plays an extremely important role in viral infections, especially chronic viral infections. In this review, I will introduce the findings to date on how IL-21 is involved in some typical viral infections and the potential of treating viral diseases with IL-21.

## 1. Introduction

In 2000, a new type 1 cytokine family receptor, named the IL-21 receptor, was discovered through a database search [1,2]. At the same time, a novel cytokine, Interleukin (IL)-21, was identified as a ligand that binds to the IL-21 receptor [1]. The IL-21 receptor forms a heterodimer together with the common cytokine receptor γ chain (γc chain) and activates STAT3, mainly as a signaling molecule by stimulation with IL-21 [3,4]. The γc chain, originally identified as the IL-2 receptor γ chain in 1992 [5], was subsequently found to be shared by other cytokines, IL-4, IL-7, IL-9, IL-15 and IL-21, in addition to IL-2. Accordingly, it is called the γc chain [6]. The γc chain was found to be the causative gene of X-linked severe combined immunodeficiency (X-SCID) [7,8]. In X-SCID patients, in addition to impaired differentiation of T cells and NK cells, B cell dysfunction is considered to be caused by impaired signal transduction from IL-21 and IL-4 receptors [9].

IL-21 is produced from activated CD4^+^ T cells and NKT cells [1] and is thought to have various functions against various immune cells [10]. Later, it was found that IL-21 is produced from two types of activated CD4^+^ T cells. One is type 17 helper T (Th17) cells, whose differentiation has been reported to be affected by IL-21 [11,12,13,14,15,16,17]. The other type of activated CD4^+^ T cells are follicular helper T (Tfh) cells, and the IL-21 produced from these cells also plays an important role in Tfh cell differentiation and germinal center formation [18,19,20]. However, subsequent studies suggest that IL-6 and ICOSL, a co-stimulatory molecule, are essential for Tfh cell differentiation, and that IL-21 may play an auxiliary role [21,22].

IL-21 is a major cytokine produced by Tfh cells along with IL-4 and CXCL13, and IL-21, IL-4 and CD40L are major stimulators for the regulation of antibody production by T-cell-dependent activated B cells in the germinal center [22]. Affinity maturation, which is based on somatic hypermutation, and class switch recombination of antibody genes in germinal centers are extremely important for the production of high-affinity antibodies and the maintenance of memory B cells, which are essential for protecting against viral infections [23]. Meanwhile, IL-21 induces apoptosis in bystander B cells activated without antigen receptor stimulation [10]. Thus, it has become clear that IL-21 has important functions, such as controlling class switching on B cells and promoting final differentiation into plasma cells (Figure 1). Studies using gene-deficient mice showed that IL-21 promotes IgG1 class switching together with IL-4, but, unlike IL-4, it was found to suppress class switching to IgE [9,24]. In addition, in an in vitro study using human B cells, IL-21 promoted class switching to IgG1 and IgG3 [25]. Subsequent studies have shown that a deficiency of STAT3, the major signaling molecule from the IL-21 receptor, causes hyper-IgE syndrome, which has an autosomal inheritance pattern [26,27]. In addition, it was found that increased IgE was observed in approximately half of patients with an IL-21 receptor deficiency, although the level of immunoglobulin itself decreased [28]. These results suggest that IL-21 enhances the class switch to IgG1 and IgG3 and suppresses the class switch to IgE in humans. In addition, in studies using IL-21 or IL-21 receptor gene-deficient mice, it was reported that IL-21 induces somatic hypermutation in B cells and is involved in the enhanced affinity of antibodies [29,30].

IL-21 is not essential for the development of CD8^+^ T cells but is known to be involved in the proliferation and activation of CD8^+^ T cells and immunological memory formation, together with other γc cytokines such as IL-7 and IL-15 [31,32,33,34,35] (Figure 2). It has also been shown that IL-21 is involved in the differentiation of CD8^+^ effector memory T cells in Listeria infection [36]. Persistent viral infection exhausts activated CD8^+^ T cells [37,38,39], but even exhausted CD8^+^ T cells retain the ability to control chronic viral infections [40,41]. Therefore, the prevention of CD8^+^ T cell exhaustion and the recovery of exhausted CD8^+^ T cells are important for the treatment of chronic viral infections and tumors. Over the past 10 years, immune checkpoint inhibition therapy that inhibits the PD-L1-PD-1 and CD28-CTLA4 system has come into focus as an immunotherapy for tumors, targeting exhausted T cells, and is actively used as the fourth treatment method for tumors. In recent years, IL-21 was reported to maintain an antiviral immune response by maintaining CD8^+^ effector T cells and suppressing CD8^+^ T cell exhaustion [42]. In the future, it is expected that a treatment method combining an immune checkpoint inhibitor and IL-21 cytokine therapy will be developed as an approach to immunotherapy for chronic viral infections and tumors.

Given the outcomes of these different studies, IL-21 is considered to be a very important cytokine that is key to the antiviral immune response in many respects, including Tfh-mediated regulation of antibody function and activation of CD8+ T cells. This review summarizes recent findings regarding the role of IL-21 in viral infection and treatment.

## 2. Vaccinia Virus and Lymphocytic Choriomeningitis Virus (LCMV) Infection

In general, IL-21 regulates the immune response to viral infections via CD4^+^ T cells, B cells and CD8^+^ T cells. While there are reports that IL-21 has an important function in the survival of CD8 + T cells against acute vaccinia virus infection [33,35], there is also a report suggesting that IL-21 is not required [43]. On the other hand, in chronic LCMV infections, IL-21 produced by CD4^+^ T cells has been shown to maintain the antiviral activity of CD8^+^ T cells and regulate CD8^+^ T cell exhaustion [43,44,45]. Signal transduction from the IL-21 receptor activates Bcl-6 and Blimp-1 via STAT3-SOCS3 together with IL-10 and promotes the differentiation of CD8^+^ memory T cells during chronic LCMV infection [46]. It has also been reported that IL-21-induced basic leucine zipper transcription factor ATF-like (BATF) induces Blimp1 expression together with IRF4 and is important for maintaining the effector function of CD8^+^ T cells [47].

In recent years, Zander et al. reported that IL-21 produced from CD4^+^ T cells promotes the maintenance of CX3CR1^+^ CD8^+^ T cells, which are considered to be CD8^+^ effector T cells to LCMV infection, and its differentiation from progenitor cells [42]. Furthermore, IL-21 suppresses the exhaustion of CD8^+^ T cells to maintain the immune response against chronic viral infections. PD-1/PD-L1 checkpoint inhibition therapy partially suppresses the appearance of exhausted T cells in patients with chronic infections and cancer, but it has also been shown that the development of exhausted CD8^+^ T cells during chronic LCMV infection could not be reverted, even by PD-1/PD-L1 checkpoint inhibition, when the production of IL-21 by CD4+ T cells was inadequate [42]. In the future, treatment that combines immune checkpoint inhibition therapy and cytokine therapy with IL-21 may be developed for chronic viral infection and cancer. Regarding the mechanism by which IL-21 inhibits differentiation into exhausted T cells, Loschinski et al. showed in an in vitro study that IL-21 induces fatty acid oxidation and oxidative phosphorylation in mitochondria to induce memory T cell differentiation and to inhibit exhausted T cell differentiation [48].

## 3. HIV Infection

In HIV patients, serum IL-21 levels decrease with a decrease in CD4^+^ T cells [49,50]. IL-21 levels were higher in patients who could control HIV compared to those who could not, and it was shown to IL-21 restore the antiviral activity of CD8^+^ T cells, similar to that of LCMV infection [49,50,51]. It has also been shown that in patients who cannot control HIV, serum IL-21 is reduced and IL-21 receptor expression on CD8^+^ T cells is increased [52]. Thus, serum IL-21 concentration and IL-21 receptor expression on CD8^+^ T cells can serve as markers for understanding the viral control status of HIV patients. Interestingly, in HIV patients with a well-controlled viral load, HIV-specific CD8^+^ T cells also produce IL-21 in a CD4^+^ T-cell-independent manner [53,54]. Because IL-21 is also known to be produced by CD8^+^ T cells in patients with other autoimmune diseases [55,56,57], its association with the pathophysiology of HIV infection is interesting.

HIV infection affects the function of B cells, such as decreased BCR reactivity and proliferative capacity, and decreased CD21 expression [58,59]. In particular, memory B cell abnormalities are prominent [60], causing insufficient antibody production in HIV patients [61]. To control HIV infection, the production of broadly neutralizing antibodies (bnAbs) is important. Twenty percent of HIV patients can produce these bnAbs [62], and these antibodies have a neutralizing ability over 70% of the currently prevalent HIV strains [63,64,65,66]. However, it is unclear why such antibodies are produced only in some affected individuals.

An important point in developing a vaccine for HIV infection is the creation of bnAbs that are widely aware of HIV. Clinical trials of HIV vaccines under several protocols have progressed; however, only the clinical trials of HIV vaccines conducted in Thailand (RV144 trial) have shown limited activity in preventing HIV infection [67]. Consequently, attention has been focused on the involvement of HIV-specific CD4^+^ T cells induced by the vaccine used in the RV144 study [68,69,70,71]. In addition, because the protocols used in clinical trials conducted so far are varied, such as vaccine strategy, administration method, boost method, etc., it is important to examine how CD4^+^ T cells react under these conditions, especially regarding the responsiveness of Tfh cells [72].

Tfh cells are the preferred target of HIV infection and are known to be a reservoir for HIV replication [73,74,75]. In contrast, Tfh cells have been reported to proliferate even under chronic HIV infection [73,76,77]. These facts suggest that HIV infection may cause abnormalities in Tfh cells, resulting in impaired helper function in B cells. Colineau’s and Graff-Dubois’ groups analyzed the splenocytes of HIV patients and found that although Tfh cells were increased in HIV patients, cytokines such as IL-4 and IL-10 and co-stimulatory molecule expression decreased, and follicular regulatory T (Tfr) cells increased. These results show that Tfh cells in HIV patients cannot achieve normal B cell activation [78,79]. On the other hand, IL-21 is also a major cytokine produced by Tfh. There are conflicting reports that IL-21 increases in HIV patients [78] and decreases or does not change in HIV patients or SIV-infected macaques [80,81]. There seems to be a difference in the amount of IL-21 produced depending on the infection status and the sample.

Morita et al. found that CD4^+^ CXCR5^+^ T cells present in human peripheral blood show a similar phenotype to Tfh cells and stimulate B cells via IL-21 [82]. In the case of human physiology, it is usually difficult to examine Tfh cell status, so analysis of the Tfh cells’ counterpart present in the periphery has been actively conducted. Locci et al. reported that PD1^+^ CXCR3^−^ CXCR5^+^ memory Tfh cells present in human peripheral blood have the most differentiation-inducing function for B cells, and their peripheral frequency correlates with bnAbs production in HIV patients [83]. Furthermore, Tfh cells, which produce IL-21 in peripheral blood, were decreased in HIV patients, and anti-HIV treatment restored peripheral Tfh cells [80]. These studies show that measuring peripheral Tfh cells in blood can estimate the status of Tfh cells in lymphoid tissues against HIV infection, providing an important indicator of HIV vaccine development. In addition, Yamamoto et al. also reported that PD1^+^ CXCR5^+^ Tfh cells are important as CD4-positive T cells were involved in bnAb production in an in vivo study using SHIV_AD8_-infected macaques [84].

Furthermore, Schultz et al. found that HIV-specific IL-21-producing CD4^+^ T cells in peripheral blood are Tfh cells circulating in peripheral blood, according to their analysis of gene expression patterns and phenotypes, and that these Tfh cells are increased in the RV144 study compared to other ineffective vaccine trials [85]. In addition, HIV-specific IL-21-producing peripheral Tfh cells correlate well with virus control in HIV patients; moreover, in poorly controlled HIV patients, peripheral Tfh cells are reprogrammed to the Th1 cell side by IL-2 signaling, resulting in decreased expression of Bcl6, CXCR5 and IL-21 [86,87]. In actual HIV infection, bnAbs are not acquired in the early stages and appear in some affected individuals in the later stages of HIV infection [88]. Although the details of the mechanism have not been clarified, it has been shown that IL-21 produced by Tfh cells plays an extremely important role in HIV-specific bnAb production [85]. If Tfh cells can properly produce IL-21 during HIV infection, bnAbs capable of controlling HIV may be produced.

In contrast, unlike bnAbs, antibodies that do not have neutralizing activity but cause innate immune activation in an Fc-dependent manner are elevated from the early stages of HIV infection and play an important role in HIV infection [89,90,91]. These antibodies are thought to confer resistance towards HIV infection through complement activation, opsonization and antibody-dependent cellular cytotoxicity (ADCC). Pusnik et al. reported that anti-HIV env-specific antibodies are produced from Blimp-1-positive B cells in an IL-21-producing CD4^+^ env-specific T cell-dependent manner, and that these antibodies are involved in Fc-dependent innate immune response activation [92]. It became clear that IL-21 produced from Tfh cells also plays an extremely important role in the production of anti-HIV antibodies that activate the innate immune response to HIV, further recognizing the importance of IL-21.

IL-21 also acts directly on HIV-infected CD4^+^ T cells and directly suppresses HIV replication by activating microRNA-29 via STAT3 [93]. This microRNA-29 activation by IL-21 shows strong natural resistance to HIV infection [94].

As described above, IL-21 resists HIV infection by activating CD8^+^ T cells, producing bnAbs and non-neutralizing innate immune-active antibodies and by directly suppressing HIV replication in CD4^+^ T cells. Animal experiments with SIV infection are being conducted to investigate the development of HIV therapy using IL-21. In SIV-infected rhesus macaques, the cytotoxic activity of SIV-specific CD8^+^ T cells decreases from the acute phase to the chronic phase, and dysbiosis and transition to the body of intestinal bacteria occurs due to a decrease in Th17 cells in the intestinal tract [95,96,97,98]. In addition, Micci et al. reported that IL-21-producing CD4^+^ T cells decreased with SIV infection, resulting in a decrease in Th17 cells [99]. Knowing that CD8^+^ T cells and Th17 cells are activated by IL-21, in vivo experiments using IL-21 have been conducted targeting these cells. In an experiment in which IL-21 was administered to rhesus macaques during the SIV chronic infection phase, the cytotoxic activity of NK cells and CD8^+^ T cells was enhanced, the anti-SIV antibody was increased, and IL-21 treatment against rhesus macaques during the SIV acute infection phase, the antiviral activity of CD8^+^ T cells and Th17 cells in the intestinal tract were maintained, but these treatments did not reduce the virus counts in either phase [100,101,102]. On the other hand, IL-21 exhibits various functions against NK cells. IL-21 promotes the proliferation of CD16^+^ NK cells and activates exhausted NK cells, while it has been also reported to limit the response of NK cells [1,31,103]. African green monkeys are non-susceptible to SIV infection, partly because their NK cells have been shown to undergo final differentiation in response to SIV infection and acquire antiviral activity [104]. Therefore, IL-21 may control the antiviral activity of NK cells. In an SIV infection study of rhesus macaques as a susceptible host, Harper et al. observed the promotion of the final differentiation of NK cells into NKG2a^low^CD16^+^ NK cells by a combination therapy of IL-21 and IFN-α in addition to antiretriviral therapy (ART), and it has been reported that the amount of SIV virus in the lymph nodes is reduced [105]. As will be described later, IL-21 has been clinically tested for malignant tumors using the activation of CD8+ T cells by IL-21, but this has not been successful due to side reactions such as liver damage. If side reactions can be controlled by limiting the method, duration and amount of IL-21 administration, stronger HIV treatment may be possible through regulation of the activity of various immune cells. In addition, by successfully administering IL-21 in parallel with vaccination, it may be possible to develop a vaccine that induces bnAbs through the appropriate activation of Tfh cells.

## 4. HBV Infection

Despite the availability of effective vaccines against HBV, there are still approximately 240 million chronic HBV infections worldwide [106]. Although HBV is often eliminated and cured in adults, infections in newborns and young people are much more likely to develop into chronic hepatitis than those in adults. The mechanism of chronicity, depending on the time of infection, is largely unknown. In a study using a mouse model of HBV infection, Publicover et al. showed that the immune system of adult mice produces more IL-21 than young mice [107]. Adult mice can eliminate viral antigens, whereas young mice cannot. IL-21 receptor-deficient mice also cannot eliminate viral antigens in the same way as young mice. Therefore, IL-21 is quite important for the anti-HBV immune response, suggesting that young patients may have reduced IL-21-mediated HBV-specific CD8 + T and B cell responsiveness. In human HBV hepatitis, there are more IL-21-producing CD4^+^ T cells in acute hepatitis than in chronic hepatitis, and the number is positively correlated with the number of HBV-specific CD8^+^ T cells [108]. In addition, virus elimination was significantly promoted in patients with elevated serum IL-21 levels 12 weeks after antiviral treatment [109]. Similar to other viral infections, IL-21 function is thought to be primarily focused on the recovery of exhausted CD8^+^ T cells [110].

Shen et al. used a mouse model to compare the immune response to persistently infectious HBV strains and non-persistently infectious HBV strains, and they discovered that IL-21 and IL-33 are induced in non-persistently infectious HBV strain-infected mice, whereas the induction does not occur in persistently infectious HBV strain-infected mice [111]. Therefore, by forcibly expressing IL-21 in hepatocytes, persistently infectious HBV strains were eliminated in a CD8^+^ T-cell-dependent manner [112,113]. It has also been shown that persistent infection with HBV can be prevented in mice in which IL-21 is transiently expressed in hepatocytes [114]. Thus, IL-21 is expected to enable the treatment and prevention of HBV infection via CD8^+^ T cells.

Appropriate antiviral antibody production is also an important antiviral defense response against HBV infections. However, with the transition to chronic hepatitis, exhaustion of CD4^+^ T cells is as prominent as the exhaustion of CD8^+^ T cells [115]. IL-21 produced from Tfh cells plays an important role in proper antibody production by B cells [10]. Interestingly, although IL-21 production from Tfh cells was significantly reduced in chronic HCV infections, antibody production by B cells was maintained [116]. This indicates that Tfh cells function IL-21-independently for the humoral immune response. Recently, Khanam et al. have reported that in chronic HBV infections, HBsAg-specific Tfh cells produce IL-27 and support the humoral immune response despite a deficiency in IL-21 production [117] (Figure 1). In the future, the difference in function between IL-21 and IL-27 in B cell differentiation will become clearer.

In contrast, IL-21 is also expected to exacerbate liver cirrhosis due to HBV infection [118]. Some patients with chronic HBV infection develop liver cirrhosis and cancer. Liver fibrosis that causes liver cirrhosis is caused by the production of α-smooth muscle actin (α-SMA) and collagen by the activation of hematopoietic stem cells (HSCs). Th17 cells are increased in liver cirrhosis [119,120] and are probably the source of increased serum IL-21 levels. Increased IL-21 may activate HSCs and exacerbate liver cirrhosis [121].

In this way, for HBV infection, IL-21 works to eliminate HBV in acute infection, but it is also involved in the transition from chronic hepatitis to liver cirrhosis, so great care should be taken when using IL-21 as a therapeutic agent [122].

## 5. HCV Infection

HCV causes chronic infection in 65–85% of affected people, and there are approximately 180 million chronic HCV patients worldwide [123]. Similar to HBV, IL-21’s function against HCV infection is thought to be mainly based on the recovery of exhausted CD8+ T cells [124]. It is clear that CD8^+^ T cells and CD4^+^ T cells play an important role in acute HCV infection [125,126].

In individuals in which neutralizing antibodies appear early in infection, spontaneous healing can be expected, whereas in individuals in which neutralizing antibodies appear late, HCV is persistently infected and induces chronic hepatitis [127,128,129]. Examining peripherally circulating Tfh cells instead of the Tfh cells in lymphoid tissues in patients with acute HCV infection, the number correlates with the amount of anti-HCV antibody, and when HCV is removed by anti-HCV drug treatment, the number of peripheral Tfh cells increases [130,131]. Salinas et al. reported that in HCV patients who heal spontaneously, peripheral Tfh cells that produce IL-21 appear early in infection, followed by anti-HCV E2 glycoprotein-specific antibodies [132].

## 6. Polyomavirus Infection

In human JC polyomavirus infection, when CD4^+^ T cells decrease due to AIDS, progressive multifocal leukoencephalopathy (PML) may develop as a brain lesion. Because the onset of PML is suppressed by JC polyomavirus-specific CD8^+^ T cells and CD4^+^ T cells, the helper function of CD4^+^ T cells against CD8^+^ T cells is considered to be important. Tissue-resident memory CD8^+^ T (CD8^+^ Trm) cells play an essential role in the immune response to polyomavirus infection according to a mouse central nervous system infection model. For the development and maintenance of CD8^+^ Trm cells, CD4^+^ T cells are required [133,134]; in particular, CXCR5^hi^ PD-1^hi^ CD4^+^ Tfh-like T cells produce IL-21 in the brain and are involved in the induction of CD8^+^ Trm cell differentiation [135] These CD4^+^ T cells express Tbet and Blimp1 and have been shown to possess both Th1 and Tfh cell traits.

## 7. SARS-CoV-2 Infection

Shortly after the first report of a new coronavirus infection in December 2019, the virus quickly spread worldwide. The acute respiratory disease caused by SARS-CoV-2, namely COVID-19, has caused many deaths worldwide [136,137]. Infection with SARS-CoV-2 activates the innate and adaptive immune system that responds to viral infections; some individuals recover without actual onset of the disease, whereas others develop severe symptoms and die. This heterogeneity is a major feature of COVID-19, and it is thought that various factors produce this heterogeneity in the innate and adaptive immune systems [138,139].

With respect to the acquired immune response during the acute and convalescent phases, many affected individuals produce IgG antibodies against the receptor-binding domain of the viral spike protein. These antibodies are important in preventing viral infections and are found to last for at least 6 months. In addition, spike protein-specific memory B cells appeared at the early stage of infection and further increased 6 months after infection [140,141,142]. Dan et al. showed that virus spike-specific CD4^+^ T cells and CD8^+^ T cells are halved in number 3–5 months after infection and that each component of each immune response shows a different trend, and there are individual differences in these trends [142].

The frequency of SARS-CoV-2-specific CD4^+^ T cells or CCR6^−^ peripheral Tfh cells in COVID-19 patients has been shown to correlate with virus-neutralizing antibody levels [143,144]. In addition, it has been reported that spike protein-specific memory B cell differentiation and antibody affinity maturation in mildly recovered COVID-19 and severely ill patients correlate well with the function of IL-21-producing CD4^+^ T cells [145]. These results show that in SARS-CoV-2 infection, similar to other viral infections, Tfh cell-induced IL-21-mediated germinal center B cell differentiation is crucial for the production of useful virus-neutralizing antibodies. As a result of analyzing B cells that differentiated in the germinal center of patients with severe COVID-19, IgG2-producing B cells that were affected by type 1 IFN at the beginning of ICU hospitalization were dominant; however, the number of IgG1- and IgA1-producing B cells affected by IL-21 and TGF-β increased as the disease progressed, and eventually, the number of IgA2-producing B cells strongly affected by TGF-β increased [146]. Furthermore, IgA2 affected by TGF-β in patients with severe COVID-19 showed unstable SARS-CoV-2 recognition. TGF-β causes pulmonary fibrosis, which is the most avoidable complication in the chronic post-inflammatory phase of SARS-CoV-2 infection. In severe COVID-19, IL-21 and TGF-β may be imbalanced, and the pathological condition is exacerbated. The development of therapeutic methods targeting these cytokines is expected in the future. A treatment method aimed at activating NK cells and CD8^+^ T cells using IL-15 for COVID-19 has been proposed [147], and Wilz argued that IL-21 should be added to this treatment [148]. If the heterogeneity of the immune response to SARS-CoV-2 infection becomes more clearly understood in the future, the development of new therapeutic methods targeting cytokines will be further promoted.

## 8. Clinical Application of IL-21

Initially, IL-21 was clinically tested for malignant melanoma, acute myeloid leukemia, metastatic renal cancer, etc., using the antitumor activity of the activation of CD8-positive T cells [149]. Some effects on tumors were observed in Phase 1 and Phase 2 clinical trials, but many were discontinued due to side reactions such as liver damage [150]. A clinical trial combining anti-EGF receptor antibody and IL-21 for metastatic colorectal cancer was also conducted, achieving stable disease in 60% of patients and observing immune cell activation [151]. However, clinical studies were discontinued when various side reactions occurred, and IL-21 was reported to be involved in the development of inflammatory bowel disease and subsequent colorectal tumors [152,153,154].

## 9. Concluding Remarks

IL-21 is a multifunctional cytokine that acts on various immune cells. In particular, the proliferation and activation of CD8^+^ T cells, the differentiation and maintenance of CD8^+^ Trm cells and the activation of exhausted CD8^+^ T cells are considered to be some of the most important functions of IL-21 as an immune response to resist viral infections, especially chronic viral infections. On the other hand, IL-21 is a major cytokine produced by Tfh cells, and it has been clarified that it has an effect of promoting optimal antibody production on B cells through class switch recombination and somatic hypermutation. Thus, IL-21 plays an extremely important role, namely the regulation of antibody production for the early convergence of acute viral infections and the recovery of exhausted T cells to resist chronic infections. Evidently, IL-21 can potentially be utilized in vaccine development for the production of optimal neutralizing antibodies and the induction of T cell immune memory. However, clinical trials of IL-21 aimed at antitumor effects have been unsuccessful, and it has been found that its clinical application is not easy. In the future, I hope that treatment and vaccine development research for viral infections using the functions of IL-21 will progress, such as the development of therapeutic strategies that limit the action to immune cells such as B cells and CD8+ T cells.

## Figures and Tables

**Figure 1 ijms-22-09521-f001:**
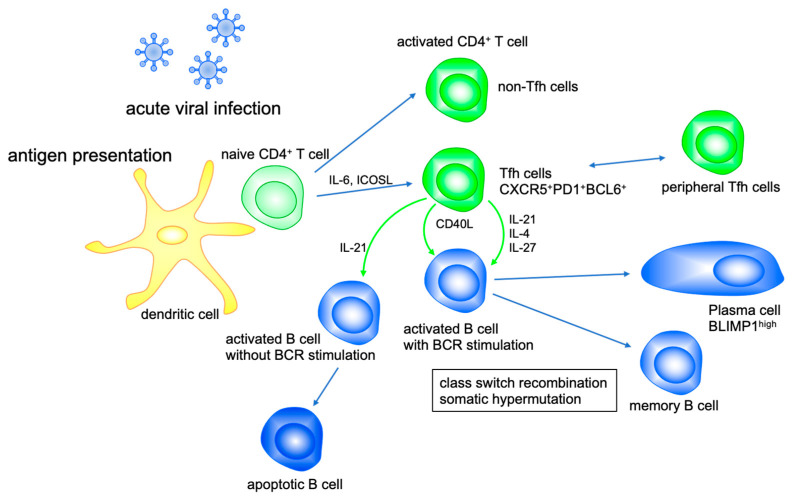
IL-21 modulates B cell function. Tfh cell activates B cell differentiation with CD40L, IL-21, IL-4 and IL-27. IL-21 promotes class switch recombination to IgG1 and IgG3 and suppresses that to IgE. IL-21 enhances somatic hypermutation and production of high-affinity antibodies by plasma cells, whereas IL-21 induces apoptosis of bystander B cells activated without antigen receptor stimulation.

**Figure 2 ijms-22-09521-f002:**
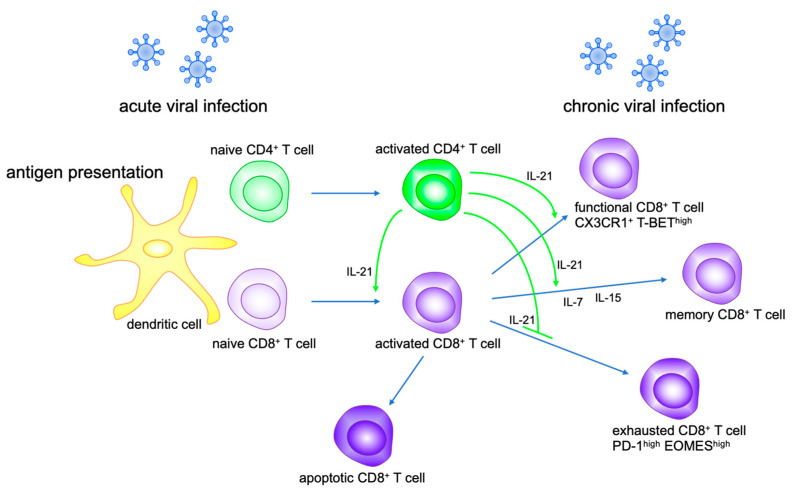
IL-21 modulates CD8^+^ T cell function. IL-21 secreted by activated CD4^+^ T cells promotes CD8^+^ T cell activation and proliferation, and differentiation to memory CD8^+^ T cells with IL-7 and IL-15. IL-21 maintains functional CD8^+^ T cells and prevents CD8^+^ T exhaustion in chronic viral infection.

## Data Availability

Not applicable.

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
