# Peer review of "Interleukin-21 in Viral Infections"

_ijms, 2021, doi:10.3390/ijms22179521_

Round 1

Reviewer 1 Report

Asao thoroughly highlights the different sources and effects of IL-21 for the humoral and cellular immunity. The manuscript is well written in terms of content. It would benefit from an English proofread to improve the clarity. I have only minor points.

Minor issues:

  • Lines 164, 165: “…similar morphology to Tfh cells…” -> similar phenotype; data was based on flow cytometry with antibody labeled cells
  • Lines 186-189: Please quote.
  • Lines 192-194: “These antibodies are thought to resist HIV infection through complement activation, opsonization, and anti-body-dependent cellular cytotoxicity (ADCC).” Please clarify the sentence; “resist” might not be appropriate (e.g. antagonize; counteract?).
  • Lines 219-221: Please clarify the sentence towards a better readability as the cited papers show promoting and inhibitory impacts for IL-21 on NK cells.
  • Lines 270, 271: “Th17 cells are increased in liver cirrhosis [119,120], resulting in increased serum IL-21 levels. IL-21 produced from Th17 cells activates HSCs and exacerbates liver cirrhosis [121].” The conclusions of both sentences might be drawn, however, they are not explicitly shown in the cited data and therefore are still hypothetic, which should be clearly visible. E.g.: Th17 cells are increased in liver cirrhosis [119,120] and are probably the source of increased serum IL-21 levels and IL-21 mediated activation of HSCs and exacerbation of liver cirrhosis [121].”
  • Lines 336, 337: “TGF-β causes pulmonary fibrosis, which is the most avoidable complication of SARS-CoV2 infection.” This might be a severe complication in the chronic post-inflammatory phase. The acute complication is a diffuse alveolar damage.

Reviewer 2 Report

The work by Asao provides a comprehensive review of IL-21 as a multifunctional cytokine that acts on various immune cells in the context of selected viral infections. The manuscript clearly summarizes the current knowledge on the cytokine's diverse roles and functions, including its production by Tfh cells, activation of CD8+ T cells, and recovery of CD8+ T cells exhausted by chronic infections.

The work is potentially interesting for the scope of the journal. However, before considering it for publications the following major and minor points should be addressed.

Major:

Asao attempts to discuss the potential therapeutic implication of IL-21. However, this aspect of the work doesn’t always emerge clearly and needs further elaboration. In this context, it would be critical for the reader to understand the current therapeutic use of IL-21 and how its targeting or therapeutic administration might be beneficial. Importantly, this would provide the rationale for further exploration of IL-21 treatment as support for host CD8+ T cell responses in viral infections cure strategies.

Minor:

Lines 47-48 – Please, specify that this is one of the main immunomodulatory effects of IL-21 and briefly explain why it is important.

Lines 97-99 – “There are reports that IL-21 has important functions, such as supporting the survival of CD8+ T cells for acute vaccinia virus infection [33,35] and, conversely, that it is not necessary” It appears that this sentence has been truncated by mistake. The authors should specify where IL-21 is “not necessary”.

Lines 99-101 – looking at figure 2, it seems that IL-21 prevents exhaustion of activated CD8+ T cells during chronic viral infections. However, the explanation in the text points towards the rescue of the antiviral activity of CD8 T cells that are already exhausted. This point should be better elaborated and, if necessary, the diagram in figure 2 must be modified accordingly.

Line 112 – Please, consider using the term “development” instead of appearance.

Lines 112-115 – Please, consider rephrasing as follows “the development of exhausted CD8+ T cells during chronic LCMV could not be reverted even by PD-1/PD-L1 checkpoint inhibition when the production of IL-21 by CD4+ T cells is inadequate”.

Lines 160-162 – “On the other hand, IL-21 is also a major cytokine produced by Tfh cells, which increases in HIV-infected persons and decreases or does not change in HIV-infected persons and SIV-infected macaques”. This statement is confusing and arguably contradicts itself. Please, consider rephrasing it.

Line 165 – Morphology is not a sensitive or specific criterion to distinguish/compare T cells subpopulations. The authors should clarify this questionable point.

Lines 192-193 – “antibodies are thought to resist HIV infection”. This sentence is confusing. It is not clear how antibodies might “resist” a viral infection. It would make much more sense if antibodies confer resistance towards a viral infection. Please, consider rephrasing this sentence.

Line 219 – “On the other hand, IL-21 exhibits various functions against NK cells”. This sentence is confusing, it appears that IL-21 “supports” NK cells instead of “exhibiting functions against NK cells”. This sentence should be rephrased for clarity.

Lines 228-230 – Please, elaborate more on how IL-21 can be used to improve vaccine development and guide new strategies and approaches in this field.

Lines 262-266 – “This indicates that Tfh cells function IL-21-independently for the humoral immune response, and in recent years, Tfh cells have been shown to produce IL-27 in chronic HBV infections to support the humoral immune response [117] (Figure 1). Differences in the functions of IL-21 and IL-27 on B cells in anti-HBV antibody production will be clarified in the future”. The meaning of these two sentences is obscure. Please, consider rephrasing them for clarity.

Reviewer 3 Report

This is a comprehensive review of IL-21 and its role in viral infections including SARS-CoV2. Reference list includes 148 articles. 

Manuscript is not easy to read. Some sentences are too long and awkward. Sometimes, transition words are not properly used. Sometimes, there is no logic connection between sentences. For example, lines 238-240: "In a study using a mouse model of HBV infection, Publicover et al. showed that the adult mouse immune system produces more IL-21, which is important for the anti-HBV immune response [107]. This low IL-21 level in young mice is thought to be caused by..."

Line 221: "African green monkeys are non-pathogenic to SIV infection...". African green monkey are non-susceptible or do show clinical signs of disease. Similarly, "a pathogenic" host in line 225.    

 Line 235: "Adult infections" need to be replaced with "adults". 

Line 294: what is the meaning of "JC" ? 
